# Focus on Therapeutic Options for Surgically Resectable Pancreatic Adenocarcinoma Based on Novel Biomarkers

**Alessandro Olivari, Virginia Agnetti and Ingrid Garajová ***

Medical Oncology Unit, Parma University Hospital, Via Gramsci 14, 43125 Parma, Italy;
alessandro.olivari@unipr.it (A.O.); virginia.agnetti@unipr.it (V.A.)
* Correspondence: ingegarajova@gmail.com; Tel.: +39-0521702660

**Abstract:** Pancreatic ductal adenocarcinoma remains associated with a poor prognosis, even when diagnosed at an early stage. Consequently, it is imperative to carefully consider the available therapeutic options and tailor them based on clinically relevant biomarkers. In our comprehensive review, we specifically concentrated on the identification of novel predictive and prognostic markers that have the potential to be integrated into multiparametric scoring systems. These scoring systems aim to accurately predict the efficacy of neoadjuvant chemotherapy in surgically resectable pancreatic cancer cases. By identifying robust predictive markers, we can enhance our ability to select patients who are most likely to benefit from neoadjuvant chemotherapy. Furthermore, the identification of prognostic markers can provide valuable insights into the overall disease trajectory and inform treatment decisions. The development of multiparametric scoring systems that incorporate these markers holds great promise for optimizing the selection of patients for neoadjuvant chemotherapy, leading to improved outcomes in resectable pancreatic neoplasia. Continued research efforts are needed to validate and refine these markers and scoring systems, ultimately advancing the field of personalized medicine in pancreatic adenocarcinoma management.

**Keywords:** neoadjuvant therapy; resectable pancreatic cancer; novel biomarkers

## 1. Introduction

According to the AIRTUM (Italian Cancer Registry), there were approximately 14,300 new diagnoses of pancreatic ductal carcinoma (PDAC) in 2020, showing an increasing trend in recent years, particularly among males. PDAC ranks as the fourth leading cause of death among men and the sixth leading cause of death among women in Italy. The five-year overall survival (OS) rate for PDAC is 8.1%, while the ten-year OS rate is 3% [1].

Non-metastatic PDAC is classified into three categories: resectable, borderline resectable, and locally advanced, based on the extent of local infiltration into adjacent structures, especially the degree of vascular involvement. Although the majority of PDAC cases are diagnosed at an unresectable stage, significant efforts have been made to address the disease at an earlier stage.

The goal is to utilize multimodality treatments, such as chemotherapy and radiotherapy, to downstage the disease, prevent micro-metastases, and enable curative surgery. However, surgical resection is only feasible in 15% to 20% of cases [2–5].

The benefits of adding a systemic treatment before surgery (neoadjuvant therapy, NAT) lies in the possibility of tumor shrinkage, to guarantee an early on delivery of chemotherapy for the treatment of radiologically invisible micro-metastatic disease and to avoid useless and potentially incurable surgery in the case of progressive disease. In the setting of resectable PDAC, neoadjuvant approaches are still controversial. According to different studies [6–8], no benefits in terms of OS were demonstrated in the addiction of NAT when compared to upfront surgery.

Therefore, in clinical practice, neoadjuvant approach for resectable PDACs is considered mainly in the presence of "high risk" features such as elevated CA19-9, large primary

tumor and/or positive regional lymph nodes or extreme pain [4,7–10], even though no precise quantitative criteria to define these features have been globally defined by the scientific community.

Unfortunately, there is a lack of clinically relevant predictive and prognostic biomarkers for this devastating disease and the challenge stands in the difficulty to foresee which patients will respond to chemotherapy with a meaningful survival benefit. At present time, both upfront radiological staging and resectability status are deployed to assist with prognostication; however, accurate biological markers in this setting are lacking [Figure 1]. Therefore, in this review, we aim to summarize several minimally invasive, cost efficient, and easily detectable biomarkers that could be used both as predictors of chemotherapy effectiveness [Table 1] and as a prognostic tool [Table 2], to shift the therapeutic decision toward the use or non-use of neoadjuvant therapy in the resectable disease [11].

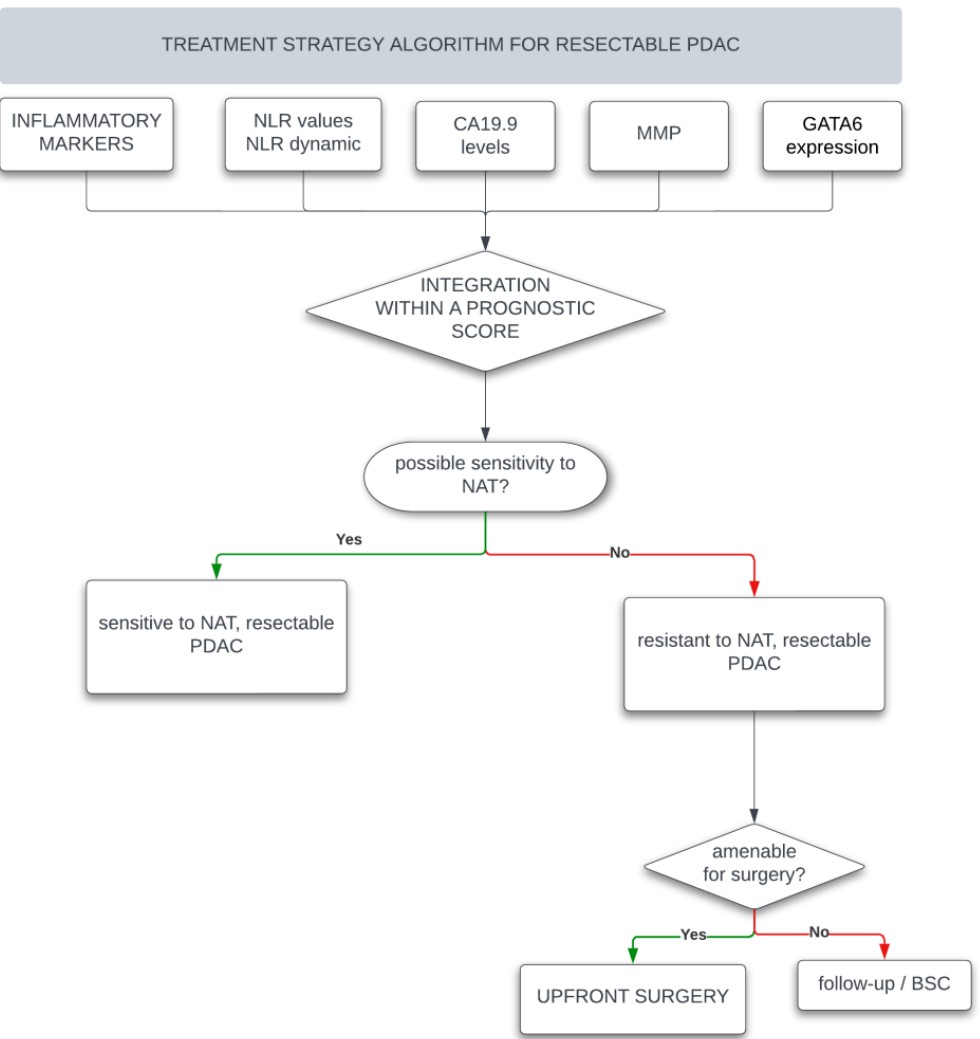

**Figure 1.** This figure summarizes the importance of tailoring treatment planning based on the most recent tumor biomarkers. It presents a flowchart designed to assist clinicians in making informed decisions, thereby minimizing the potential for unnecessary overtreatment of patients.

**Table 1.** Predictive biomarkers—This table serves as a comprehensive reference, listing various tumor biomarkers that have been extensively studied and found to exhibit a strong correlation with therapy response and/or tumor shrinkage. The inclusion of these biomarkers aims to provide a thorough overview of the molecular indicators that have shown potential in predicting treatment effectiveness and tumor size reduction.

| Biomarker | % Tumor Regression ($p$ Value) | DFS in Months ($p$ Value) | Median OS in Months ($p$ Value) | Value Cutoff | Reference |
|---|---|---|---|---|---|
| PLR | $p = 0.03$ | NA | NA | 150 | Maloney et al., 2023 [11] |
| Post-treatment CA19.9 | Higher likelihood of pMR for lower CA19.9 values ($p < 0.01$) | NA | NA | 37 U/mL | Perri et al., 2021 [12] |
| post-NACT CA19.9 | NA | NA | 35.2 (above cutoff) vs. 19.4 (below cutoff) ($p = 0.038$) | 91.8 U/mL | Heger et al., 2020 [13] |
| Perineural invasion (present) | NA | $p = 0.016$ | $p = 0.006$ | NA | Redegalli et al., 2022 [14] |
| Lymph node ratio | NA | $p < 0.0001$ | $p < 0.0001$ | NA | Redegalli et al., 2022 [14] |
| Stroma-to-neoplasia ratio | NA | $p = 0.021$ | $p = 0.002$ | NA | Redegalli et al., 2022 [14] |
| PINC | NA | $p < 0.0002$ | $p < 0.0001$ | $\geq 0.599$ | Redegalli et al., 2022 [14] |
| NLR | $p = 0.012$ | NA | NA | NA | Murakami et al., 2022 [15] |
| NLR pre-chemo + ΔNLR | NA | $p = 0.006$ | $p = 0.002$ | NA | Silva et al., 2022 [16] |
| IL2Ra | $p = 0.045$ | NA | NA | NA | Chopra et al., 2021 [17] |

NACT = Neoadjuvant chemotherapy; PLR = platelet monocyte ratio; pMR = pathological major response (<5% of viable cancer cells); PINC: pathological tumor regression scoring; MMP-7 = metalloproteinase 7.

**Table 2.** Prognostic biomarkers—This table provides an extensive compilation of tumor biomarkers that have been extensively studied and found to be significantly linked to disease progression and overall survival in various clinical contexts. The inclusion of these biomarkers aims to enhance our understanding of their prognostic value and potential implications for patient outcomes.

| Biomarker | Median RFS in Months ($p$ Value) | Median OS in Months ($p$ Value) | Median DFS in Months ($p$ Value) | Value Cutoff | Reference |
|---|---|---|---|---|---|
| NLR | NA | 13.0 above cutoff, 32.4 below cutoff ($p = 0.001$) | NA | 5 | Maloney et al., 2023 [11] |
| NLR | NA | 18.8 below cutoff vs. 10.6 above cutoff ($p < 0.001$) | NA | 3.69 | Xu et al., 2021 [18] |
| PLR | NA | 20.20 below cutoff vs. 16.50 above cutoff ($p = 0.031$) | NA | 141.7 | Xu et al., 2021 [18] |
| CA19.9 | NA | 14.2 above cutoff, 19.4 below cutoff ($p = 0.004$) | NA | 1000 U/mL | Xu et al., 2021 [18] |
| mGPS | NA | $p = 0.028$ | NA | NA | Maloney et al., 2023 [11] |

**Table 2.** *Cont.*

| Biomarker | Median RFS in Months (*p* Value) | Median OS in Months (*p* Value) | Median DFS in Months (*p* Value) | Value Cutoff | Reference |
|---|---|---|---|---|---|
| SIRIpost-neoadjuvant | NA | NA | *p* = 0.030 | 0.8710 | Kim et al., 2022 [19] |
| SIRIquotient | NA | *p* = 0.037 | NA | 0.9516 | Kim et al., 2022 [19] |
| SII | NA | *p* = 0.05 | NA | 900 | Murthy et al., 2020 [20] |
| S100A2 | NA | *p* < 0.001 | NA | NA | Dreyer et al., 2020 [21] |
| S100A4 | NA | *p* < 0.001 | NA | NA | Dreyer et al., 2020 [21] |
| T-CD9 | *p* = 0.007 | NA | NA | NA | Ahn X et al., 2022 [22] |
| S-CD9 | NA | *p* = 0.005 | NA | NA | Ahn X et al., 2022 [22] |
| MMP-7 | 37.3 (negative value) vs. 13.8 (positive value), *p* = 0.03 | 38.2 (negative value) vs. 27.6 (positive value), *p* = 0.049 | NA | IHC positivity | Shoucair et al., 2022 [23] |

LMR—Lymphocyte-to-monocyte ratio, PLR—platelet-to-lymphocyte ratio, NLR—neutrophil-to-lymphocyte ratio, mGPS—modified Glasgow performance scale (CRP > 10 mg/L and albumin < 35 g/L combined, with a higher CRP (>10 mg/L) and lower albumin (<35 g/L) indicating a worse prognosis). SIRI (systemic inflammatory response index) = nNeutrophils × nMonocytes/nLymphocytes. SIRI quotient = SIRIpostneoadjuvant/SIRIpreneoadjuvant, SII = systemic inflammatory index (SII = P × [N/L]) T-CD9 (tumor-CD9), S-CD9 (stroma-CD9).

## 2. Tumor Immune Microenvironment

Inflammation plays a crucial role in the development, progression, and metastasis of various adult malignancies, including pancreatic ductal adenocarcinoma (PDAC) [22,24]. Among the circulating inflammatory cells, neutrophils have been extensively studied in different tumor types and are associated with more aggressive disease. Studies have shown that a higher number of neutrophils correlates with unfavorable outcomes in PDAC.

In this setting, the CXCR2 chemokine receptor, responsible for leukocyte chemotaxis and inflammation regulation, has been linked to an increased tumor size and a worse prognosis in PDAC [7,8,25,26].

PDAC triggers a robust local and systemic inflammatory response through cytokine-mediated proliferation of circulating plasma cells (cPCs) and fibroblasts, hindering effective drug delivery to the tumor [20,27]. Additionally, the tumor microenvironment (TME) of PDAC is characterized by the proliferation of pancreatic stellate cells (PSCs) and fibroblasts, as well as the infiltration of myeloid-derived suppressor cells (MDSCs) and tumor-associated macrophages (TAMs). These cells suppress the activation of immune cells, impede tumor-specific immunosurveillance, and promote tumor growth and metastasis [28]. The peritumoral desmoplastic reaction further impedes angiogenesis, compromising the delivery of chemotherapeutic drugs to the tumor and leading to chemoresistance [29]. Targeting inflammatory mediators has therefore been proposed as a strategy to enhance chemosensitivity and improve therapy response in PDAC [30,31].

A better understanding of the tumor microenvironment and its adaptation following chemotherapy is crucial for comprehensively characterizing PDAC. This knowledge may pave the way for the development of novel treatment strategies, addressing the complex interplay between inflammation, TME, and therapy response.

## 3. Prognostically Significant Tumor Immune Biomarkers

In recent times, there has been a growing focus on the discovery of immune biomarkers and gene signatures that can accurately predict the response to immunotherapy and provide

valuable insights into the tumor immune environment (TME). These findings not only have predictive significance in determining the effectiveness of immunotherapy but also play a prognostic role in forecasting disease progression and mortality.

In this regard, the modulation of DNA methylation, histone acetylation, and histone methylation has shown potential in enhancing the efficacy of immune checkpoint inhibitors. Additionally, it can induce a transformation of tumor cell clusters from those associated with a poorer prognosis, a heightened expression of oncogenes, and limited immune cell infiltration to clusters that exhibit an increased sensitivity to standard chemotherapy regimens and a more robust immunological microenvironment [32–38].

Furthermore, extensive research has been conducted on numerous enzymes, receptors, transporters, and proteins, including ADAMTS12, TLR3 (Toll-like receptor 3), GLUT1 (glucose transporter 1), SQLE (squalene epoxidase), SHCBP1 (protein binding to the SH2 domain of Src collagen homolog), RNF43 (E3 ubiquitin ligase), and IFI27 (interferon alpha-inducible protein). These investigations have aimed to explore their associations with tumorigenesis, cancer progression, the modulation of tumor immune cell infiltration, the expression of immune checkpoints, and the upregulation of immunosuppressive genes. These molecules hold potential as future biomarkers for diagnosing and prognosing pancreatic adenocarcinoma, as well as serving as therapeutic targets for tumor immunotherapy [39–44].

Moreover, scoring systems based on the measurement of metabolites such as N6-methyladenosine (m6A) could serve as independent prognostic factors for predicting the response to immunotherapy and assessing modifications in immune cell infiltration, including Tregs, CD8, and CD4 [45–47].

Furthermore, extensive investigations have focused on gene signatures that encode mutated tumor antigens and their connection to immune-activated phenotypes and survival outcomes. Consequently, significant efforts have been devoted to developing gene prognostic models and risk scores that can independently predict survival and provide further insights into the tumor immune microenvironment [32,48].

Finally, targeting the KRAS gene mutation has demonstrated its potential impact on the outcomes of pancreatic cancer patients, instilling hope for the development of future drugs to treat heavily pretreated cases [49–51].

## 4. Predictive and Prognostic Role of Neutrophil to Lymphocyte Ratio

Several studies have aimed to explore the correlation between the inflammatory response in the tumor microenvironment and the prognosis of pancreatic ductal adenocarcinoma (PDAC). It has been revealed that PDAC patients with tumor neutrophil infiltration tend to have a worse prognosis compared to those with lymphocytic infiltration, who may have better survival chances [52]. Consequently, numerous studies have focused on investigating the interplay between inflammatory marker levels, such as the neutrophil-to-lymphocyte ratio (NLR), platelet-to-lymphocyte ratio (PLR), and lymphocyte-to-monocyte ratio (LMR), and their impact on prognosis. These markers have been associated with cancer growth, migration, and invasion promotion [11].

Furthermore, research conducted by Maloney et al. [11] has indicated that a higher NLR predicts a shorter overall survival, while a higher PLR is correlated with an increased tumor viability during surgery in patients receiving neoadjuvant chemotherapy. Other studies have observed that a high NLR before neoadjuvant treatment can predict the success of operability in resectable/borderline resectable PDAC [53–56]. Conversely, a low preoperative NLR has been associated with a worse overall survival and a disease-free survival [15]. The dynamics of NLR and its potential to predict chemotherapy response have also been studied in preclinical models. These studies have shown that treatments that reduce NLR not only significantly downstage tumor burden and metastatic growth, but also increase the presence of tumor-infiltrating CD8+ T cells, thereby dampening fibroblast polarization and chemo-resistance signaling pathways [16].

However, contrasting results have been reported, with some studies demonstrating that although NLR variations during neoadjuvant treatment were significant, they were not associated with pathological response, overall survival, or disease-free survival [57,58]. Moving forward, there is a need for a more precise standardization of NLR evaluation to enhance its clinical utility in predicting treatment outcomes and prognosis.

## 5. Predictive and Prognostic Role of Inflammatory Markers

In the pursuit of meaningful predictive and prognostic biomarkers, there has been a growing interest in investigating the role of inflammatory mediators, such as C-reactive protein (CRP) and albumin, in predicting survival outcomes in pancreatic ductal adeno-carcinoma (PDAC) [59,60]. Specifically, the modified Glasgow prognostic score (mGPS) has been proposed, incorporating both albumin and CRP levels, as a potential prognostic marker in gastrointestinal cancers, including PDAC. Multiple studies have demonstrated the accuracy of the mGPS in determining overall survival (OS), with lower albumin levels and higher CRP levels associated with a worse prognosis [61].

Furthermore, efforts have been made to elucidate the role of inflammatory markers and their interplay in tumorigenesis and disease progression. Chopra et al. [17] identified IL-2Ra levels as a significant predictor of response to neoadjuvant therapy. They developed a decision tree combining IL-2R$\alpha$, IL-12p40, IL-6, and IL-8 to predict response to neoadjuvant treatment, achieving high sensitivity and specificity. Additionally, they found that an upregulated inflammatory status was associated with a poorer response to neoadjuvant therapy.

Other studies have explored the potential of the systemic inflammatory index (SII) as a prognostic and predictive tool for tumor response to neoadjuvant therapy. SII, calculated by multiplying neutrophil and monocyte counts and dividing them by the lymphocyte count, was found to be an independent negative predictor of OS, surpassing the predictive value of the neutrophil-to-lymphocyte ratio (NLR). SII levels were directly associated with CA19.9 levels [20]. Kim et al. [19] investigated the inflammatory response index (SIRI), calculated by combining neutrophil and monocyte counts and dividing them by the lymphocyte count, as a potential tool to predict the efficacy of neoadjuvant therapy and disease progression. Higher post-neoadjuvant SIRI levels were associated with worse disease-free survival and a higher risk of recurrence after surgery. They also demonstrated that a higher SIRI quotient, calculated as the ratio of SIRI values before and after neoadjuvant therapy, was linked to poorer overall survival. These findings highlight the potential of inflammatory markers in predicting treatment response and disease progression in PDAC.

## 6. Predictive and Prognostic Role of Tumor Secreted Biomarkers

One potential approach for identifying pancreatic ductal adenocarcinoma (PDAC) patients who would likely benefit from neoadjuvant therapy (NAT) is through the selection of biomarkers, such as GATA6 expression [62,63]. Tumors expressing GATA6, classified as the classical subtype, tend to exhibit higher responses to FOLFIRINOX, while those with the basal-like subtype have low or no GATA6 expression, leading to worse outcomes [64].

In this regard, the role of CA19-9 has been found to be directly linked to overall survival (OS) [18]. A study by Liu et al. [65] indicated that fluctuations in CA19-9 levels during neoadjuvant therapy can predict whether patients may benefit from additional adjuvant therapy. Moreover, higher levels of CA19-9 before and after neoadjuvant therapy were associated with a lower likelihood of tumor response and a higher risk of tumor persistence. However, the correlation between CA19-9 levels and pathologic response is not fully established [12]. Additionally, Heger et al. [13] focused on the variation of CA19-9 levels before and after NAT with FOLFIRINOX and demonstrated that post-neoadjuvant values could more accurately predict resectability and survival among PDAC patients compared to the progressive evolution of CA19-9 levels.

Another study [66] emphasized the importance of pre-operative levels of CA19-9 and NLR (neutrophil-to-lymphocyte ratio) as key factors in predicting no early recurrence

in resectable PDAC patients. Among the various tumor-secreted biomarkers considered, both tumor and stromal CD9 values emerged as more precise prognostic indicators for patients receiving NAT, as their expression was strengthened by the selective pressure of chemotherapy [67].

In addition to these biomarkers, Wada et al. [68] found that the levels of phospho-choline, carnitine, and glutathione were higher in patients who received chemoradiotherapy in the neoadjuvant setting and were associated with a better disease-free survival (DFS) compared to the control group without treatment. Furthermore, in this setting, S100A2 and S100A4 were associated with poor survival and a high risk of early recurrence after NAT and before surgery [21].

Lastly, metalloproteinases (MMPs) are known to play a role in the pathogenesis and progression of PDAC [69]. Specifically, MMP-7 is involved in upregulating mitogen-activated protein kinase-dependent pathways and is associated with an increased invasiveness of pancreatic tumors by stimulating EGFR-mediated pathways [70–72]. Shoucair et al. [23] suggested studying MMP-7 levels on fine-needle aspiration (FNA) specimens at the time of diagnosis as a potential prognostic and predictive marker for pathologic response to NAT.

## 7. Combined Biomarkers Scores

Extensive research aimed at identifying key factors for better patient stratification in different prognostic groups has resulted in the proposal of several scoring systems. One such system is the PANAMA score [73], which combines tumor size, positive nodes, R status, and CA19-9 levels. This risk score has demonstrated superior accuracy in predicting overall survival (OS) and recurrence risk after surgery compared to the AJCC staging system. Another scoring system, the BACAP score [74], has also been proposed, showing a correlation with OS based on clinical parameters such as venous/arterial thrombosis, performance status, pain, weight loss, tumor topography, and maximal tumor size.

Moreover, perineural invasion and lymph node ratio (the ratio of positive nodes to the total number of sampled lymph nodes) have been found to be associated with a shorter OS and disease-free survival (DFS), while an elevated stroma-to-neoplasia ratio (a qualitative evaluation of the area covered by these two components) has been linked to longer OS and DFS. These findings have been translated into a comprehensive tumor regression scoring system known as PINC, which can be easily applied in clinical practice [14,23].

## 8. Conclusions

Currently, the standard treatment approach for resectable pancreatic ductal adenocarcinoma (PDAC) involves surgery, followed by adjuvant therapy, while the role of neoadjuvant therapy in this context remains to be clarified.

Therefore, the identification of easily accessible predictive and prognostic biomarkers is crucial for guiding therapeutic decisions. Numerous studies have highlighted the intricate connection between the tumor microenvironment and tumor behavior, aiding in the understanding of which patients may benefit from neoadjuvant therapy and which may not due to chemoresistance. MMP-7 has shown promise as a potential predictive marker for neoadjuvant therapy response. The neutrophil-to-lymphocyte ratio (NLR) and its fluctuations during treatment also play a promising role. Several studies have indicated that higher pre-neoadjuvant therapy NLR levels are associated with a poorer response, while monitoring dynamic changes in NLR during treatment can provide valuable insights into the effectiveness of neoadjuvant therapy. We believe that the development of combined scoring systems could serve as a significant guide for determining the suitability of neoadjuvant therapy in surgically resectable pancreatic cancer, enabling the selection of patients who are likely to benefit from this approach in the future.

**Author Contributions:** Conceptualization, A.O. and V.A.; validation, I.G.; writing—original draft preparation, A.O. and V.A. writing—review and editing, A.O., V.A. and I.G.; supervision, I.G.; All authors have read and agreed to the published version of the manuscript.

**Funding:** This research received no external funding.

**Conflicts of Interest:** The authors declare no conflict of interest.

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
