# Peer review of "Focus on Therapeutic Options for Surgically Resectable Pancreatic Adenocarcinoma Based on Novel Biomarkers"

_curroncol, doi:10.3390/curroncol30070475_

Round 1

Reviewer 1 Report

This is an interesting review of novel predictive and prognostic biomarkers for surgically resectable pancreatic ductal adenocarcinoma in the context of the debate of the efficacy of neoadjuvant theraphy (NAT).

The review is well written with the biomarkers grouped by type.

I only have a couple of notes concerning Figure&Tables.

Figure and Tables should be cited in the text and a suitable legend should be added. Specifically, the authors should explain whether Figure 1 simply represents a summary of the data reviewed or whether it explains their own suggestions how decision making should be conducted.

As for the Tables 1 and 2, titled Predictive and Prognostic Biomarkers, respectively, it is not clear why they include both biomarkers and scoring algorithms.  Moreover, in the conclusion, MMP-7 is highlighted as one of the most promising predictive marker for NAT response, however is not included in either tables. Criteria for inclusions in the table could be explain in the legend.

Minor corrections:

In section 2., line 82, the short term for Circulating Plasma Cells should be cPCs (rather than PSC).

NLR in the title of section 3., line 98,  should be written in full.

In section 5., line 191, the bracket should be removed.

In Figure 1, ‘sensible’ to NAT, should be changed with ‘sensitive’ to NAT.

Author Response

Dear Reviewer,

Thank you for your valuable feedback and suggestions on our article. We sincerely appreciate the time you have dedicated to reviewing our work.

We are pleased to inform you that we have made significant revisions to address the points you raised. In particular, we have provided additional details regarding the tables and figures, ensuring that the data presented is more comprehensive and easier to interpret for the readers.

Furthermore, as per your request, we have included MMP (Matrix Metalloproteinase) values in our updated version.

Thank you once again for your support.

Best regards,

Alessandro Olivari, MD

Reviewer 2 Report

The manuscript reads well, and mainly focuses on the tumor immune microenvironment, but doesn't mention many other markers besides this. The title, therefore, is somewhat misleading: it should be on novel IMMUNE biomarkers. 

There are a bunch of reviews already referring to these issues in Pubmed, some of which should probably be cited, that would make the review more complete... and more informative. Its also a bit on the short side. A bit more substance wouldnt hurt! There are hundreds of potential (bio)markers discussed in the literature. On the one hand, this alone would justify a focus on a specific area, like immune markers. But on the other hand, some of the most relevant markers discussed in the field actually are related to the immune microenvironment, immune evasion, and failure of pretty much all immune checkpoint inhibitor therapies in PDAC. 

One aspect entirely missing is the formation of a fibrotic or desmoplastic tumor microenvironment, and the role of cancer-associated fibroblasts in immune evasion. Also, the role of antigen-presenting and inflammatory CAFs (apCAFs, iCAFs) versus other, fibrosis-generating CAFs (liky myCAFs). 

Then, I think the authors should maybe address a few of these rather lo-hanging fruits (markers, mutation patterns, signaling etc) that are likely to have an impact on overall survival AND immune staatus of tumors; and which would be nice to read in a truly comprehensive review on PDAC immunotherapies: 

Programmed Death-Ligand 1 (PD-L1):. PD-L1 expression in pancreatic adenocarcinoma has been studied as a natural and logical predictor of response to immune checkpoint inhibitors, such as pembrolizumab or nivolumab.

Certain gene signatures, such as interferon-gamma (IFN-γ) signature, have been associated with improved response to immunotherapy in pancreatic adenocarcinomas, in addition to the cell-based markers discussed in the manuscript.

Tumor mutational burden (TMB): Higher TMB has been associated with increased immunogenicity, meaning that the tumor is more likely to be recognized by the immune system. TMB is being explored as a potential biomarker for response to immunotherapies in pancreatic adenocarcinoma.

Microsatellite instability (MSI) or DNA mismatch repair deficiency (dMMR): Tumors with MSI or dMMR may respond to immune checkpoint inhibitors, such as pembrolizumab or nivolumab, which can enhance the body's immune response against cancer cells.

BRCA1/2 mutations: Patients with germline BRCA1 or BRCA2 mutations may be candidates for targeted therapies, such as poly ADP-ribose polymerase (PARP) inhibitors like olaparib or rucaparib. But they may also predict for immune checkpoint inhibitors, or other, targeted drugs.

HER2/neu overexpression: HER2/neu-positive pancreatic cancers may benefit from targeted therapies like trastuzumab, but also from immuno therapies

KRAS mutations: Although targeted therapies specifically against KRAS mutations are still under development, various approaches are being investigated in clinical trials. There are some papers on the putative predictive marker of KRAS mutations for certain therapies and immune status. 

Some of the more recent papers worth considering for including are mentioned below: 

Li N, Jia X, Wang Z, Wang K, Qu Z, Chi D, Sun Z, Jiang J, Cui Y, Wang C.

Characterization of anoikis-based molecular heterogeneity in pancreatic cancer

and pancreatic neuroendocrine tumor and its association with tumor immune

microenvironment and metabolic remodeling. Front Endocrinol (Lausanne). 2023 May

10;14:1153909. doi: 10.3389/fendo.2023.1153909. PMID: 37234801; PMCID:

PMC10206226.

Zou X, Guo Y, Mo Z. TLR3 serves as a novel diagnostic and prognostic

biomarker and is closely correlated with immune microenvironment in three types

of cancer. Front Genet. 2022 Nov 7;13:905988. doi: 10.3389/fgene.2022.905988.

PMID: 36419829; PMCID: PMC9676367.

Ye Y, Zhao Q, Wu Y, Wang G, Huang Y, Sun W, Zhang M. Construction of a

cancer-associated fibroblasts-related long non-coding RNA signature to predict

prognosis and immune landscape in pancreatic adenocarcinoma. Front Genet. 2022

Sep 23;13:989719. doi: 10.3389/fgene.2022.989719. PMID: 36212154; PMCID:

PMC9538573.

Kawakubo K, Castillo CF, Liss AS. Epigenetic regulation of pancreatic

adenocarcinoma in the era of cancer immunotherapy. J Gastroenterol. 2022

Nov;57(11):819-826. doi: 10.1007/s00535-022-01915-2. Epub 2022 Sep 1. Erratum

in: J Gastroenterol. 2022 Sep 17;: PMID: 36048239; PMCID: PMC9596544.

Kato S, Fujiwara Y, Hong DS. Targeting <i>KRAS</i>: Crossroads of Signaling

and Immune Inhibition. J Immunother Precis Oncol. 2022 Aug 17;5(3):68-78. doi:

10.36401/JIPO-22-5. PMID: 36034582; PMCID: PMC9390702.

Dai L, Mugaanyi J, Cai X, Lu C, Lu C. Pancreatic adenocarcinoma associated

immune-gene signature as a novo risk factor for clinical prognosis prediction in

hepatocellular carcinoma. Sci Rep. 2022 Jul 13;12(1):11944. doi:

10.1038/s41598-022-16155-w. PMID: 35831362; PMCID: PMC9279485.

Wang N, Zhu L, Wang L, Shen Z, Huang X. Identification of SHCBP1 as a

potential biomarker involving diagnosis, prognosis, and tumor immune

microenvironment across multiple cancers. Comput Struct Biotechnol J. 2022 Jun

18;20:3106-3119. doi: 10.1016/j.csbj.2022.06.039. PMID: 35782736; PMCID:

PMC9233189.

You W, Ke J, Chen Y, Cai Z, Huang ZP, Hu P, Wu X. SQLE, A Key Enzyme in

Cholesterol Metabolism, Correlates With Tumor Immune Infiltration and

Immunotherapy Outcome of Pancreatic Adenocarcinoma. Front Immunol. 2022 May

26;13:864244. doi: 10.3389/fimmu.2022.864244. PMID: 35720314; PMCID: PMC9204319.

Li F, He C, Yao H, Liang W, Ye X, Ruan J, Lin L, Zou J, Zhou S, Huang Y, Li

Y, Chen S, Huang K, Lian G, Chen S. GLUT1 Regulates the Tumor Immune

Microenvironment and Promotes Tumor Metastasis in Pancreatic Adenocarcinoma via

ncRNA-mediated Network. J Cancer. 2022 May 13;13(8):2540-2558. doi:

10.7150/jca.72161. PMID: 35711842; PMCID: PMC9174867.

Xu H, Yin L, Xu Q, Xiang J, Xu R. N6-methyladenosine methylation

modification patterns reveal immune profiling in pancreatic adenocarcinoma.

Cancer Cell Int. 2022 May 23;22(1):199. doi: 10.1186/s12935-022-02614-x. PMID:

35606813; PMCID: PMC9125922.

Song C, Chen J, Zhang C, Dong D. An Integrated Pan-Cancer Analysis of

ADAMTS12 and Its Potential Implications in Pancreatic Adenocarcinoma. Front

Oncol. 2022 Feb 23;12:849717. doi: 10.3389/fonc.2022.849717. PMID: 35280819;

PMCID: PMC8904364.

Liu Y, Li G, Yang Y, Lu Z, Wang T, Wang X, Liu J. Analysis of

N6-Methyladenosine Modification Patterns and Tumor Immune Microenvironment in

Pancreatic Adenocarcinoma. Front Genet. 2022 Jan 3;12:752025. doi:

10.3389/fgene.2021.752025. PMID: 35046996; PMCID: PMC8762218.

Hosein AN, Dangol G, Okumura T, Roszik J, Rajapakshe K, Siemann M, Zaid M,

Ghosh B, Monberg M, Guerrero PA, Singhi A, Haymaker CL, Clevers H, Abou-Elkacem

L, Woermann SM, Maitra A. Loss of Rnf43 Accelerates Kras-Mediated Neoplasia and

Remodels the Tumor Immune Microenvironment in Pancreatic Adenocarcinoma.

Gastroenterology. 2022 Apr;162(4):1303-1318.e18. doi:

10.1053/j.gastro.2021.12.273. 

Huang S, Zhao J, Song J, Li Y, Zuo R, Sa Y, Ma Z, OuYang H. Interferon

alpha-inducible protein 27 (IFI27) is a prognostic marker for pancreatic cancer

based on comprehensive bioinformatics analysis. Bioengineered. 2021

Dec;12(1):8515-8528. doi: 10.1080/21655979.2021.1985858. PMID: 34592906; PMCID:

PMC8806992.

Mao M, Ling H, Lin Y, Chen Y, Xu B, Zheng R. Construction and Validation of

an Immune-Based Prognostic Model for Pancreatic Adenocarcinoma Based on Public

Databases. Front Genet. 2021 Jul 14;12:702102. doi: 10.3389/fgene.2021.702102.

PMID: 34335699; PMCID: PMC8318842.

Wang L, Zhang S, Li H, Xu Y, Wu Q, Shen J, Li T, Xu Y. Quantification of m6A

RNA methylation modulators pattern as a potential biomarker for prognosis and

associated with tumor immune microenvironment of pancreatic adenocarcinoma. BMC

Cancer. 2021 Jul 31;21(1):876. doi: 10.1186/s12885-021-08550-9. PMID: 34332578;

PMCID: PMC8325189.

Wang C, Shi M, Zhang L, Ji J, Xie R, Wu C, Guo X, Yang Y, Zhou W, Peng C,

Zhang H, Yuan F, Zhang J. Identification of KRAS G12V associated clonal

neoantigens and immune microenvironment in long-term survival of pancreatic

adenocarcinoma. Cancer Immunol Immunother. 2022 Feb;71(2):491-504. doi:

10.1007/s00262-021-03012-4. Epub 2021 Jul 13. PMID: 34255132; PMCID: PMC8783870.

Huang X, Zhang G, Tang T, Liang T. Identification of tumor antigens and

immune subtypes of pancreatic adenocarcinoma for mRNA vaccine development. Mol

Cancer. 2021 Mar 1;20(1):44. doi: 10.1186/s12943-021-01310-0. PMID: 33648511;

PMCID: PMC7917175.

Saung MT, Zheng L. Adding combination immunotherapy consisting of cancer

vaccine, anti-PD-1 and anti-CSF1R antibodies to gemcitabine improves anti-tumor

efficacy in murine model of pancreatic ductal adenocarcinoma. Ann Pancreat

Cancer. 2019 Dec;2:21. doi: 10.21037/apc.2019.11.01. PMID: 32405624; PMCID:

PMC7220030.

Li J, Byrne KT, Yan F, Yamazoe T, Chen Z, Baslan T, Richman LP, Lin JH, Sun

YH, Rech AJ, Balli D, Hay CA, Sela Y, Merrell AJ, Liudahl SM, Gordon N, Norgard

RJ, Yuan S, Yu S, Chao T, Ye S, Eisinger-Mathason TSK, Faryabi RB, Tobias JW,

Lowe SW, Coussens LM, Wherry EJ, Vonderheide RH, Stanger BZ. Tumor Cell-

Intrinsic Factors Underlie Heterogeneity of Immune Cell Infiltration and

Response to Immunotherapy. Immunity. 2018 Jul 17;49(1):178-193.e7. doi:

10.1016/j.immuni.2018.06.006. 

Author Response

Dear Reviewer,

Thank you for reviewing our article and providing your feedback. We acknowledge your suggestions and have made the necessary corrections accordingly.

We would like to address your comment regarding the need for additional references to support the discussion on immune biomarkers in pancreatic adenocarcinoma. In response, we have included another paragraph in the revised manuscript, citing the papers you mentioned. These additional references strengthen the discussion and provide further evidence on the role of immune biomarkers in this context.

We appreciate your critical evaluation of our work, which has helped us enhance the scientific rigor and validity of the article.

Kind regards,

Alessandro Olivari,MD

Round 2

Reviewer 2 Report

The authors have picked up some of the suggestions from the reviewers and added sections to cover some of these comments. This is a relatively short article, therefore, I think its appropriate that they haven't tried to add EVERYTHING that was suggested. 

Then, the authors added a large number of new references, which is very appropriate (in my opinion) expanding the scope of the manuscript considerably. 

And since it is a relatively short manuscript, I would advise against adding a number of new stories, or storylines, to it... its currently quite focused and that's a good thing. 

only the usual typos that can be fixed in production of the article